# The Pattern Recognition Receptor FLS2 Can Shape the Arabidopsis Rhizosphere Microbiome β-Diversity but Not EFR1 and CERK1

**DOI:** 10.3390/plants11101323

**Published:** 2022-05-17

**Authors:** Jose P. Fonseca, Venkatachalam Lakshmanan, Clarissa Boschiero, Kirankumar S. Mysore

**Affiliations:** 1The Noble Research Institute, Ardmore, OK 73401, USA; vlakshmanan@loambio.com (V.L.); clarissaboschi@yahoo.com (C.B.); 2Institute for Agricultural Biosciences, Oklahoma State University, Ardmore, OK 73401, USA; 3Department of Biochemistry and Molecular Biology, Oklahoma State University, Stillwater, OK 74078, USA

**Keywords:** 16s, ITS, microbiome, rhizosphere, PRR, PAMPs

## Abstract

Pathogen associated molecular pattern (PAMP) triggered immunity (PTI) is the first line of plant defense. We hypothesized that the absence of pattern recognition receptors (PRRs) in plants could influence the rhizosphere microbiome. Here, we report sequencing of the 16S ribosomal RNA gene and the fungal ribosomal RNA internal transcribed spacer region of rhizosphere DNA from three *Arabidopsis* PRR mutants involved in plant innate immunity (*efr1*, *fls2*, and *cerk1*). We conducted experiments in a growth chamber using native soil from the Red River Farm (Terral, OK, USA) to detect microbial community shifts in the rhizosphere that may occur in the absence of PRR receptors compared to wild-type (WT; Col-0) plants. No difference in the α-diversity of the rhizosphere microbial population was observed between the PRR mutants tested and the WT. Plant host genotype had a significant impact in bacterial β-diversity only between the *fls2* mutant and the WT. Surprisingly, no significant changes in fungal β-diversity were observed between the PRR mutants and WT, although we observed an increase in relative abundance for the cup fungi (*Pezizaceae*) in the *cerk1* mutant. This finding suggests that the FLS2 receptor can modulate the rhizosphere-associated microbiome β-diversity and expands the list of current known genotypes that can modulate the rhizosphere microbiota.

## 1. Introduction

The effect of host genotype in the root-associated microbiome is still not well understood, although many studies have laid out the foundations for a better understanding of how plant genetic components regulate the plant-associated microbiome [1,2,3,4]. A large study using hundreds of *Arabidopsis thaliana* plants from eight different accessions, grown in two different soil types, and developmental stages showed that despite changes in soil composition and plant growth stage, host genotype remained a preponderant factor influencing endophytic compartment (EC) microbiota [1]. Another study in *Arabidopsis* using different knockout mutants involved in hormone-mediated defense responses, mainly the salicylic acid pathway like *cpr1*, *cpr5*, *pad4*, *sid2*, and *snc1* mutants, detected shifts in the bacterial microbiome from the EC at several taxonomic levels like phylum and family [4]. The bacterial microbiome in barley (*Hordeum vulgare*) was analyzed using different genotypes of wild and domesticated barley. The effect of host plant genotype in shaping the root-associated microbiome was small but significant [5]. The abundance of several bacterial taxa in root endophytic communities varied across different plant phosphate-starvation response mutants, reinforcing the role of genes involved in stress responses as key drivers of microbiome changes besides soil composition [6]. 

A different conclusion was reached in a study using hundreds of wild accessions of *Arabidopsis*; the authors showed that host plant genotype (different *Arabidopsis* ecotypes) mainly affected the Pseudomonadaceae specifically, whilst having almost no effect on the majority of the microbiome [3]. This study also showed a correlation of the growth of the rhizosphere dwelling beneficial bacteria *Pseudomonas fluorescens* in different *Arabidopsis* accessions and the effect on overall plant growth and health. 

A recent study showed that genotype-alone effect could not be observed using mutants impairing different metabolic pathways [2]. In this study, the authors tested mutants impairing different pathways such as flavonoids using the *chalcone isomerase* (*tt5*) mutant, tryptophan-derived defense metabolites using the *cytochrome P450 79B2* and *cytochrome P450 79B3* (*cyp79Bb2/b3*) mutants, methionine-derived defense metabolite mutant (*myb28*) and coumarin biosynthesis knockout mutant (*feruloyl-coenzyme A ortho-hydroxylase 1*; *f6′h1*). The only mutant genotype that caused shifts in microbial community composition was *f6′h1* under iron deficiency conditions [2]. Other studies also suggest a limited effect of plant genotype in the rhizosphere microbiome from wheat and soybean, respectively [7,8]. Interestingly, one study in grapevine found a more pronounced effect of host genotype in the rhizosphere microbiome [9]. A better understanding of how plant host genotype may influence its surrounding microbiomes could be important in the future to help engineering stress tolerant crops and improving the plant adaptation to their immediate surrounding environments.

The first line of defense in plants, also known as pathogen-associated molecular pattern (PAMP) triggered immunity (PTI), can recognize PAMPs through specialized pattern recognition receptors (PRRs) and activate a defense signaling cascade [10,11]. Lack of PAMP surveillance leads to enhanced disease susceptibility in plants. Several PRRs have been identified and characterized in *Arabidopsis*. Well known examples of PRRs in PAMP recognition are the receptor-like kinases (RLKs) such as flagellin sensitive 2 (FLS2), ef-tu receptor (EFR), and lysin motif-containing chitin elicitor receptor kinase 1 (CERK1) that can recognize microbial features such as bacterial flagellin protein domain flg22, elongation factor EF-TU (elf18 peptide), and fungal cell wall polysaccharide chitin, respectively [12,13,14,15]. 

Although a correlation was previously established [2,4,5] between *Arabidopsis* host genotype and root microbial population using mutants involved in plant hormone defense responses and phosphate starvation responses, no study has profiled changes in microbial communities from the rhizosphere, modulated by PRRs involved in PTI. Here, we analyzed for the first time the composition and abundance of the rhizosphere microbiome for *Arabidopsis* PRR receptor mutants *fls2*, *efr1*, and *cerk1* through bacterial 16S rRNA and fungal ribosomal RNA internal transcribed spacer (ITS) sequencing.

## 2. Results

### 2.1. Taxonomic Composition of Bacterial and Fungal Rhizosphere Communities

#### 2.1.1. Bacterial Relative Abundance

Relative abundance of the most represented bacterial phyla was determined for the rhizosphere (Figure 1 and Appendix A). The most abundant phylum across most samples was Proteobacteria (*cerk1*–29%, WT–27%, *efr1*–34%), with the exception of the *fls2* mutant, which was Cyanobacteria (30%). The second most abundant phylum was Cyanobacteria for *cerk1* and WT (16% for both), while for the *efr1* and *fls2* mutants, it was Planctomycetes and Proteobacteria (14% and 23%, respectively) (Figure 1).

At the bacterial family level for rhizosphere samples, we observed the predominance of Burkholderiaceae across most genotypes (*cerk1*–13%, WT–9%, *efr1*–15%), with the exception of the *fls2* mutant, which was Phormidiaceae (17%) (Figure 2 and Appendix A). The second most abundant taxa at the family level was Pirellulaceae (*cerk1*–7%, WT–7%) with the exception of the *fls2* and *efr1* mutants, which was Burkholderiaceae (13%) and Phormidiaceae (9%), respectively (Figure 2). Increased abundance of Phormidiaceae was observed for the *fls2* mutant in comparison to the WT (17% vs. 4%, respectively). In contrast, some taxa such as Haliangiaceae (0.7% vs. 3%), Chitinophagaceae (0.5% vs. 3%), Anaerolineae (2% vs. 4%), and Rhizobiaceae (2% vs. 3%) were less abundant in the *fls2* mutant compared to the WT (Figure 2). 

#### 2.1.2. Fungal Relative Abundance

At the phylum level, fungal population for the rhizosphere was almost entirely divided between Ascomycota and Basidiomycota with Ascomycota being the dominant phylum (*cerk1*–78%, WT–77%, *efr1*–74%, *fls2*–73%) and Basidiomycota the second dominant phylum (*cerk1*–13%, WT–10%, *efr1*–18%, *fls2*–19%) (Appendix A).

At the rhizosphere class level, we observed the Ascomycete class Dothideomycetes as the most abundant (*cerk1*–40%, WT–40%, *efr1*–32%, *fls2*–39%) (Figure 3). The second most abundant class present in the rhizosphere was Sordariomycetes (*cerk1*–18%, WT–28%, *efr1*–26%, *fls2*–23%). Interestingly, Pezizomycetes was highly enriched in the *cerk1* mutant (17%) but not in other genotypes (1–6%).

### 2.2. The Pezizaceae Family Is Highly Enriched in the cerk1 Mutant and All Other Taxonomic Ranks

Species richness based on relative abundance for fungi rhizosphere at the family level indicated a predominance of Didymellaceae for the *efr1* and *fls2* mutants (*efr1*–17%, *fls2*–20%), while for the WT and *cerk1* mutant, it was Nectriaceae and Pezizaceae (19% and 17%, respectively) (Figure 4, Appendix A). For the *efr1* and *fls2* mutants, the second most enriched family was Nectriaceae (*efr1*–12%, *fls2*–15%), while for the WT and *cerk1* mutant, it was Didymellaceae (16%). Interestingly, Pezizaceae to which the cup fungi (mushrooms that grows in the shape of a cup) belongs to was highly enriched in the *cerk1* mutant (17%) but not in other genotypes (1–6%), in a similar way to what was observed at the class level (Pezizomycetes; Figure 3). In contrast, several taxa such as Nectriaceae (11% vs. 19%), Chaetomiaceae (5% vs. 7%), and Pleosporaceae (3% vs. 4%) were less abundant in the *cerk1* mutant compared to the WT (Figure 4).

### 2.3. Compositional Differences in the Microbial Community between Genotypes

The diversity of microbial community in the rhizosphere can be analyzed by looking at the α-diversity and β-diversity. Alpha-diversity is the observed species richness or taxa number of a given sample as well as their relative abundance. Beta-diversity on the other hand is the variability in community composition or how dissimilar two communities can be [16]. Alpha-diversity measurements based on Faith’s phylogenetic diversity for bacterial rhizosphere samples (Figure 5) indicated that bacterial diversity did not vary significantly between genotypes (*p*-value < 0.05) (Table 1). 

In contrary, the weighted unifrac β-diversity measurement showed that the *fls2* mutant is significantly different compared with the WT rhizosphere samples (*p*-value < 0.05) (Figure 6A and Table 1). The differences between the *fls2* mutant and WT control for β-diversity can be seen in the principal coordinate analysis (PCoA) of the bacterial community (Figure 6A) with no overlapping samples between WT and the *fls2* mutant. No significant difference was observed in both α-diversity and β-diversity between tested genotypes for fungal population in the rhizosphere (Figure 6B and Table 1).

## 3. Discussion

In order to better understand how the plant genotype can influence the rhizosphere microbiota, some studies succeeded in identifying specific genes from host plants that can cause specific shifts in rhizosphere microbial diversity [2,4,5]. Plants recognize microbes via conserved signatures often referred to as PAMPs. Several PRRs have been identified that can perceive PAMPs to induce PTI. In this work, we identified the rhizosphere microbiome from key *Arabidopsis* PRR mutants (*fls2*, *efr1*, and *cerk1*) and WT (Col-0) plants grown on native soil from the Red River Farm (OK, USA) to look for shifts in microbial abundance and diversity.

Since PRR receptors are involved in the first line of defense by recognition of PAMPs in plants and triggering plant defense signaling pathway [10,17], we hypothesized that the absence of PRR receptors could affect the composition and/or diversity of the rhizosphere microbiome. Surprisingly, we did not see a dramatic difference in composition and/or diversity of both bacterial and fungal population in the PRR mutants tested when compared to WT control. Overall, the composition of the bacterial and fungal communities from the rhizosphere was in agreement with previous studies using different soil types from different locations within the USA [4,8]. 

The predominant bacterial phylum for rhizosphere samples was Proteobacteria across all genotypes followed by Planctomycetes. Fungal rhizospheric samples were dominated by Ascomycetes and Basidiomycetes at the phylum level as expected [18,19]. The predominant fungal classes found in the rhizosphere were the Dothideomycetes and Sordariomycetes across all genotypes. At the family level for fungi, we identified the predominant family taxa as being Didymellaceae and Nectriaceae for the rhizosphere with the exception for the Pezizaceae, which was highly enriched in the *cerk1* mutant alone (17%) but not in other genotypes (1–6%). Interestingly, important plant pathogens such as sclerotinia, *Peziza*
*sclerotiorum* Lib. (the causal agent of cotton soft rot disease), and *Phymatotrichopsis omnivora* (the causal agent of cotton/alfalfa root rot disease) belong to this taxon [20,21]. The lysin motif (LysM) receptor-like kinase CERK1 is a well-studied example of a PRR perceiving peptide PAMP. CERK1 is the *Arabidopsis* receptor for the fungal cell wall polysaccharide chitin [15,22] present in fungi but not in plants. Mutation in the *CERK1* gene (*cerk1* mutant used in this work) can block downstream chitin-responsive genes resulting in increased susceptibility to fungal pathogens [22]. Another mutation in the *CERK1* gene have been identified (*cerk1-4)*, that can cause the opposite effect, enhanced defense responses [23], but this mutant was not used in this work. CERK1 has also been shown to be able to perceive bacterial PAMP signals and *Pseudomonas syringae* effectors in *Arabidopsis* [24,25]. 

Our data did not indicate a significant impact of PAMP mutants in the rhizopshere α-diversity compared to WT plants for both bacteria and fungi. For β-diversity, on the other hand, we observed a significant difference in composition between the *fls2* mutant and WT (*p*-value = 0.015) for bacterial samples. No significant changes were observed for other samples. For fungal β-diversity, we found no significant changes. Interestingly, only the *fls2* mutant significantly drove shifts in the rhizosphere microbiota β-diversity. These data suggest that the FLS2 receptor could, mechanistically, modulate plant responses to the root-associated bacteria. 

In the past ten years, several studies investigated the interplay between plant genotype and the root-associated microbiome [2,3,4,6]. These studies arrived at similar, although independent, conclusions about the specific role of plant genotype in shaping the root-associated microbiome. Other factors—for example, soil type, geographical location, and presence/absence of roots—were pointed as major drivers of shifts in microbiome composition [19,26,27]. Indeed, Bulgarelli et al. [5] reported that plant genotype only contributed around 6% of the total rhizosphere microbiome composition. Perhaps not all genes can drive significant shifts in the root-associated microbiome equally. However, the emerging picture from this work and previous rhizosphere microbiome studies is that species richness (e.g., taxa abundance), and β-diversity can be influenced by plant genotypes [28] in a specific manner [3]. It is worth noting that understanding which plant host genes and the precise mechanisms by which they can drive shifts in microbial community could be important to help engineering stress tolerant crop plants in the future. One could speculate that the host plant genotype alone could change how plants can effectively communicate with beneficial or pathogenic microbes, and thus potentially impacting how it adapts to their immediate surrounding environments.

## 4. Materials and Methods

### 4.1. Plant and Soil Material Preparation

Non-cultivated native soil was collected from the top 30 cm of earth from the Noble Research Institute Red River Farm (Ardmore, OK, USA), which is part of the Noble Research Institute campus. Soil physicochemical properties defined this soil mainly as a loamy sand type of soil (80% sand, 14% silt and 5% Clay, Appendix A).

*Arabidopsis thaliana* Col-0, *fls2*, *cerk1*, and *efr1* homozygous mutant seeds (obtained from Dr. Cyril Zipfel, University of Zurich) were sterilized in a laminar flow hood by incubation of seeds in 100% ethanol for 1 min, followed by a 10 min incubation in 40% bleach and 0.1% Triton-X100. Seeds were then washed 4× in sterile dH_2_O and transplanted with sterile tips to ½ strength Murashige & Skoog (MS) plates with 1% sucrose. Seeds were kept in a dark cold room (4 °C) for three days and germinated in the growth chamber at 22 °C under 12 h light/12 h dark photoperiod for two weeks. Plants were then aseptically transplanted to Red River Farm soil (non-autoclaved) for another 4–5 weeks until harvesting root tissue. Plants were only watered with autoclaved, distilled water to avoid local microbial water contamination. 

### 4.2. Sampling Procedure

After six weeks of growth, Arabidopsis roots were harvested and soil attached to the roots, the rhizosphere, was isolated for each genotype in five replicates (n = 5). To isolate rhizosphere from *Arabidopsis* roots, we manually removed loose soil from roots with the help of sterile forceps and gloves until only soil surrounding the roots remained. We then placed roots with attached rhizospheric soil into clean 50 mL plastic tubes containing 15 mL of sterile 1× phosphate buffer (10 mM PBS) pH 7.4 (1.44 g of NaHPO_4_, 0.24 g of KH_2_PO_4_, 0.2 g KCl, and 8 g NaCl in 1 L of dH_2_O). Next, we vortexed roots in buffer for 10 s and filtered turbid solution using a sterile nylon mesh to filter solid debris and aggregates. The debris-free solution was centrifuged at maximum speed for 6 min and supernatant was discarded to obtain the final rhizosphere pellet. All samples were stored immediately at −80 °C for DNA extraction.

### 4.3. DNA Extraction

Total microbial community DNA was obtained from 0.3 g of rhizospheric soil using the Power Soil DNA Isolation Kit (MoBio Laboratories, Carlsbad, CA, USA) using the manufacturer’s instructions.

### 4.4. 16S rDNA and ITS Sequencing

We used primers 515F (GTGCCAGCMGCCGCGGTAA) and 806R (GGACTACHVGGGTWTCTAAT) to amplify the V4 region of the 16S rRNA gene [29] for each sample. For ITS amplicon, we used primers 5.8S (AACTTTYRRCAAYGGATCWCT) and ITS4 (AGCCTCCGCTTATTGATATGCTTAART) [30]. PCR reactions were performed with around 20 ng template DNA and overall DNA quality was estimated by electrophoresis in 1% agarose gels in 1× TAE buffer (40 mM Tris, pH 8.3; 20 mM acetic acid; 1 mM EDTA), stained with SyberSafe dye (Thermo Fisher Scientific, Waltham, MA, USA) and quantified using Nanodrop (Thermo Fischer Scientific NanoDrop 2000, Waltham, MA, USA). PCR reactions were done with KAPA HiFi HotStart PCR, with PCR blockers peptide nucleic acid (PNA) clamps mPNA and pPNA [31] to reduce contamination from mitochondrial and plastidial 16S organelle DNA. The PCR program was as follows: 95 °C for 3 min, 30 cycles of: 98 °C for 20 s, 75 °C for 10 s (PNA clamping), 60 °C for 15 s, 72 °C for 15 s, and final extension at 72 °C for 3 min. Each PCR reaction was prepared to a final volume of 25 µL: 5× Kappa HiFi buffer (12 µL); primer 515F (1 µL); primer 806R (1 µL); mixed PNAs (1 µL each) water (5 µL); 4 µL DNA to a final concentration of 20 ng/ µL.

Reactions were purified using AMPure XP magnetic beads (Agencourt Biosciences, Beverly, MA) according to manufacturer’s instructions. Reactions were quantified using a Qubit 3.0 fluorometer (Life Technologies, Carlsbad, CA, USA) and equimolar amounts of each sample pooled to a final 4 nM concentration for sequencing. Purified eluted samples were barcoded using Nextera XT DNA Library Prep Kit (Illumina, San Diego, CA, USA). PCR conditions for index PCR were as follows (72 °C for 3 min, 95 °C for 3 min, 8 cycles of: 95 °C for 10 s, 55 °C for 30 s, 72 °C for 30 s, and final extension at 72 °C for 5 min). Index PCR reactions were prepared to a final volume of 50 µL: Index1 (5 µL); Index2 (5 µL); Kapa mix (12.5 µL); dH_2_O (10 µL); DNA (5 µL). Sequencing was performed by on an Illumina MiSeq instrument (Illumina, San Diego, CA, USA).

### 4.5. Sequence Processing for 16S and ITS Amplicons

Taxonomic assignment was performed using the software Quantitative Insights into Microbial Ecology 2 (QIIME2) [32]. Quality filtering steps for trimming adaptors and primers from demultiplexed paired reads was performed with Cutadapt v.2.1 [33]. The Cutadapt processed reads were further processed using the DADA2 algorithm as a QIIME2 plugin [34] to filter low-quality and chimera errors and generate a final amplicon sequence variants (ASV) table and corresponding taxonomic table. Taxonomy to the ASVs was assigned using a naïve Bayes algorithm implemented in the q2-feature-classifier prefitted on the SILVA 132 database (v.13.8) V4–V5 16S region of 16S gene [35], and UNITE database v.7.2 (UNITE community, 2017) for the ITS. ASVs classified as mitochondria and chloroplast were discarded from the ASV table. Nonprevalent ASVs defined as ASVs not present in at least 10% of our samples and low abundance ASVs that had less than 100 read counts across all the samples were filtered out. After processing, the taxonomy was reassigned using filtered ASVs.

The 16S and ITS library pool data was rarefied to 19,000 and 17,000 sequence reads, respectively, for the bacterial and fungal α- and β-diversity analyses to prevent potential bias caused by different sequencing depths. To assess α-diversity of bacterial communities, observed operational taxonomic units (OTUs), Shannon diversity, phylogenetic diversity (Faith PD), and evenness were calculated across different genotypes. To evaluate β-diversity indices, unweighted (for community membership) and weighted (for community structure) UniFrac distances [36] were calculated using QIIME2 pipeline. Differences in overall bacterial and fungal community membership and structure were visualized using principal coordinate analysis (PCoA) plots of these dissimilarity matrices. Differences between treatment groups were tested statistically using permutational multivariate analysis of variance (PERMANOVA; adonis function in the vegan package).

For the ITS sequencing, a total of 2,429,263 ITS raw reads were sequenced (average of 60,000 reads per sample). For bacteria, a total of 2,018,242 raw reads were generated, and after denoising and filtering, a total of 1,334,080 16S rRNA reads were used in the downstream analysis (average of 72,000 reads per sample) (Appendix A).

## 5. Conclusions

In summary, this study demonstrated no significant impact of different PRR-mutants (compromised in immune response) on α-diversity of the rhizosphere microbial community. However, we detected a significant change in the β-diversity of rhizosphere-associated microbiome composition for the *fls2* mutant compared to the WT control. The relative abundance of specific community members also varied across different taxonomic groups like the fungal taxa Pezizaceae that was consistently enriched in the *cerk1* mutant compared to the WT control and other PRR mutants.

## Figures and Tables

**Figure 1 plants-11-01323-f001:**
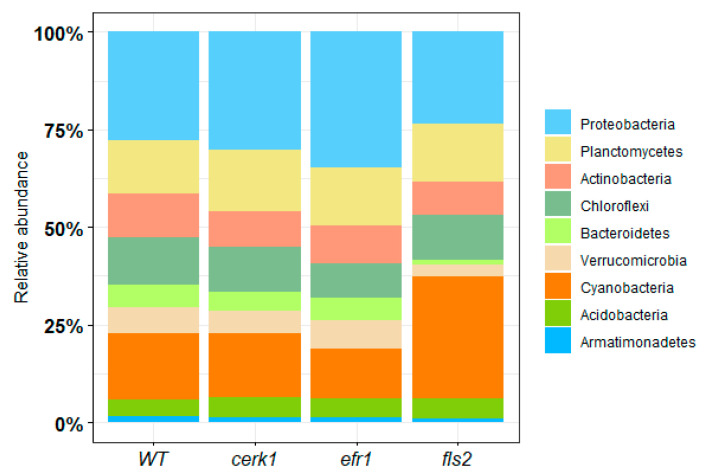
Relative abundance of the most represented enriched bacterial phyla from the rhizosphere of *Arabidopsis* PAMP mutants (*cerk1*, *efr1*, and *fls2*) and the wild-type (WT; Col-0) grown in native soil from the Red River Farm (OK, USA). Bacterial phyla with <1% abundance were not included.

**Figure 2 plants-11-01323-f002:**
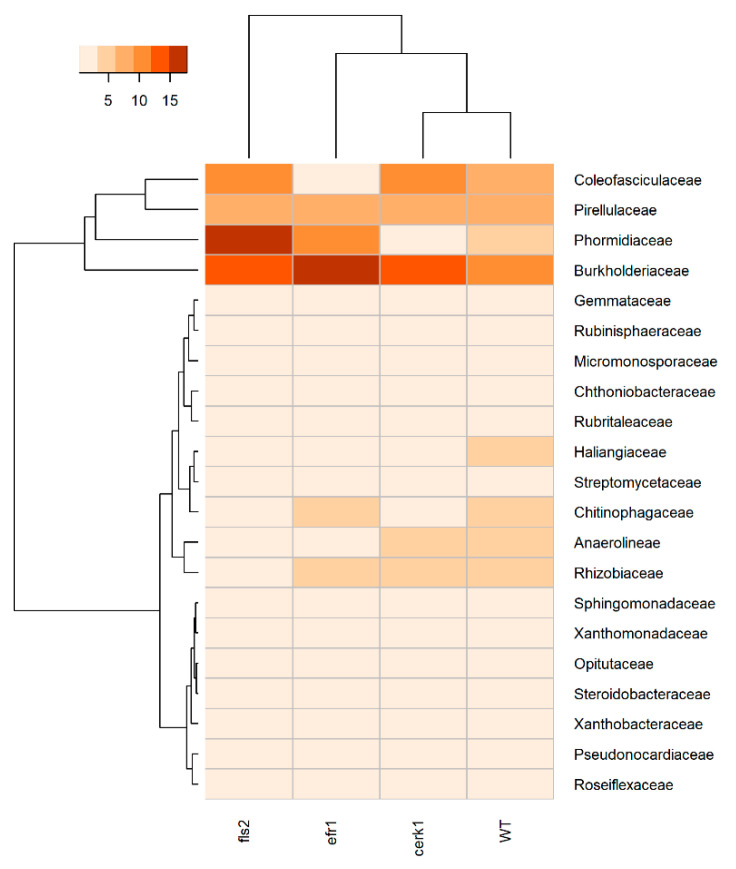
Heatmap displaying bacterial relative abundance at the family level for three different *Arabidopsis* PAMP mutants (*cerk1*, *efr1*, and *fls2*) and the wild-type (WT; Col-0) grown in native soil from the Red River Farm (OK, USA). Heatmap is color-coded based on relative abundance values. Less enriched samples are shown in light orange/pink while dark orange/red represents highly enriched samples. Samples with similar microbial profiles were recursively grouped together in branches of a dendrogram. Families with <1% were not included. Hierarchical clustering of distances using taxonomy relative abundance data were generated using Euclidean distance calculation to obtain a matrix and dendrogram construction using R.

**Figure 3 plants-11-01323-f003:**
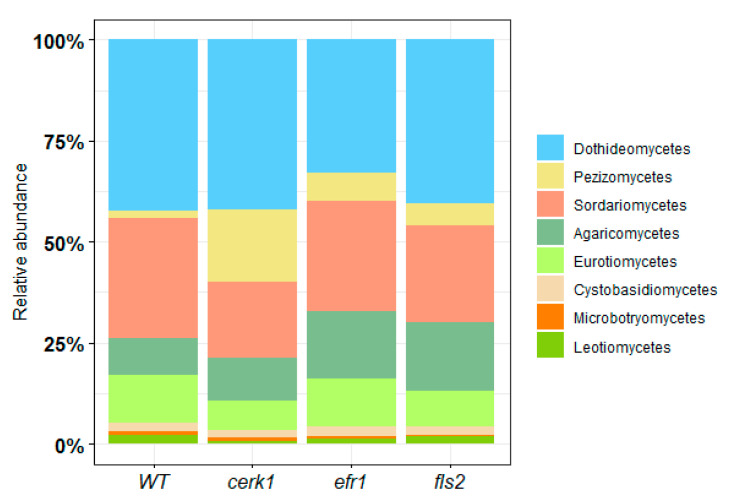
Relative abundance of the most represented enriched fungal class from the rhizosphere of *Arabidopsis* PAMP mutants (*cerk1*, *efr1*, and *fls2*) and the wild-type (WT; Col-0) grown in native soil from the Red River Farm (OK, USA). Fungal classes that were <1% abundant were not included.

**Figure 4 plants-11-01323-f004:**
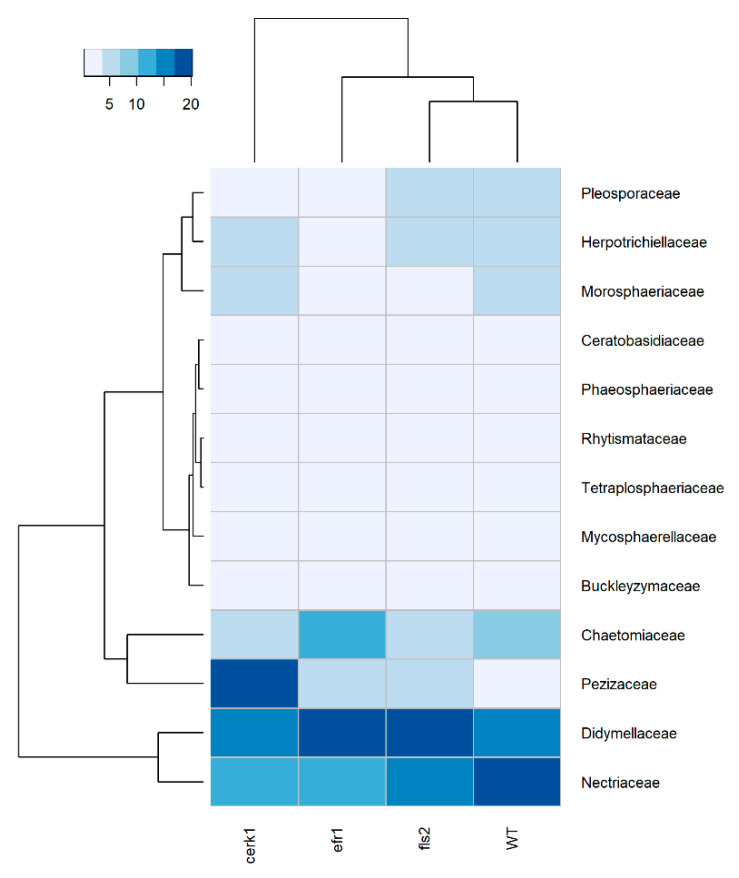
Heatmap displaying fungi relative abundance at the family level from the rhizosphere of *Arabidopsis* PAMP mutants (*cerk1*, *efr1*, and *fls2*) and the wild-type (WT; Col-0) grown in native soil from the Red River Farm (OK, USA). Heatmap is color-coded based on relative abundance values. Less enriched samples are shown in light blue/grey while dark blue represents highly enriched samples. Samples with similar microbial profiles were recursively grouped together in branches of a dendrogram at the top. Families with <1% abundance were not included. Hierarchical clustering of distances using taxonomy relative abundance data were generated using Euclidean distance calculation to obtain a matrix and dendrogram construction using R.

**Figure 5 plants-11-01323-f005:**
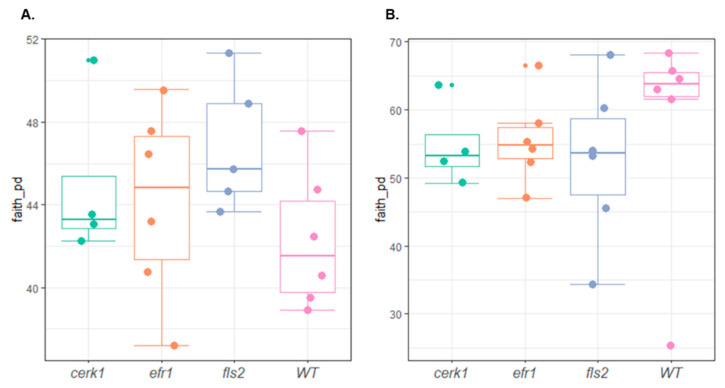
Histogram depicting α-diversity of the bacterial (**A**) and fungal (**B**) communities from rhizosphere of *Arabidopsis* PAMP mutants (*cerk1*, *efr1*, and *fls2*) and the wild-type (WT; Col-0) grown in the Red River Farm soil (Table 1).

**Figure 6 plants-11-01323-f006:**
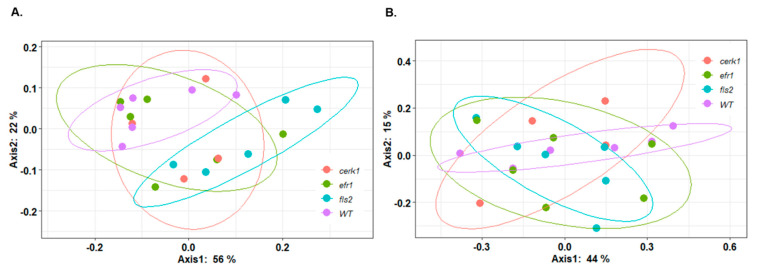
Principal coordinate analysis (PCoA) of microbial β-diversities based on weighted unifrac taxa abundances for bacterial (**A**) and fungal (**B**) samples obtained from rhizosphere of *Arabidopsis* PAMP mutants (*cerk1*, *efr1*, and *fls2*) and the wild-type (WT; Col-0). Distances between circles on the ordination plot reflect dissimilarities in bacterial and fungal community structure. Different colors denote different genotypes used.

**Table 1 plants-11-01323-t001:** *p*-values of bacterial and fungal samples compared across different genotypes.

Microbial Community	Genotypes	α-Diversity (Faith)	β-Diversity (Weighted Unifrac)
Bacterial	*cerk1* and *efr1*	0.83	0.75
	*cerk1* and *fls2*	0.14	0.16
	*cerk1* and WT	0.28	0.32
	*efr1* and *fls2*	0.36	0.07
	*efr1* and WT	0.33	0.29
	*fls2* and WT	0.06	0.01
Fungi	*cerk1* and *efr1*	0.66	0.84
	*cerk1* and *fls2*	1	0.76
	*cerk1* and WT	0.20	0.50
	*efr1* and *fls2*	0.63	0.79
	*efr1* and WT	0.20	0.64
	*fls2* and WT	0.20	0.73

## Data Availability

The data presented in this study are available upon request from the corresponding author.

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
