# Peer review of "The Pattern Recognition Receptor FLS2 Can Shape the Arabidopsis Rhizosphere Microbiome β-Diversity but Not EFR1 and CERK1"

_plants, 2022, doi:10.3390/plants11101323_

Round 1

Reviewer 1 Report

The work of Fonseca et al. reported results obtained from the analysis of rhizosphere microbiome (bacteria and fungi) of Arabidopsis mutants defective in pathogens recognition receptors discussing the possible involvement of genotype in shaping the rhizosphere microbial community. The work is undoubtedly interesting, as the interactions between roots, plants and soil microorganisms are still largely to be understood. The results of the research identified one of the three receptors under study (FLS2) as a possible modulator of the beta diversity of the rhizosphere microbiome. The work is well written and the results are well discussed. Minor points are the following: -Material and methods in Plants are at the end of the manuscript-please modify according to “guide to the authors”- LINE 113 please insert in the paragraph title “16S rDNA”…and ITS…- LINE 165 ITS raw reads were sequenced,…please insert the number of reads that were used…LINE 169-176 this is not a part of results, please move it at the beginning of the Discussion LINE 176-180 this is part of Mat and methods section, please move it on the right section; LINE 183 please change “top” with most “represented”…also in the other parts of the manuscript; LINE 192 please change “taxa for family” with “taxa at family level”

Author Response

Reviewer 1.

The work of Fonseca et al. reported results obtained from the analysis of rhizosphere microbiome (bacteria and fungi) of Arabidopsis mutants defective in pathogens recognition receptors discussing the possible involvement of genotype in shaping the rhizosphere microbial community. The work is undoubtedly interesting, as the interactions between roots, plants and soil microorganisms are still largely to be understood. The results of the research identified one of the three receptors under study (FLS2) as a possible modulator of the beta diversity of the rhizosphere microbiome. The work is well written and the results are well discussed. Minor points are the following:

-Material and methods in Plants are at the end of the manuscript-please modify according to “guide to the authors”  done

- LINE 113 please insert in the paragraph title “16S rDNA”…and ITS… done

- LINE 165 ITS raw reads were sequenced,…please insert the number of reads that were used…done

It’s already described in the manuscript text: “For the ITS sequencing a total of 2,429,263 ITS raw reads were sequenced”

LINE 169-176 this is not a part of results, please move it at the beginning of the Discussion; done

 LINE 176-180 this is part of Mat and methods section, please move it on the right section; done

LINE 183 please change “top” with “most represented”…also in the other parts of the manuscript;  done

LINE 192 please change “taxa for family” with “taxa at family level”  done

Reviewer 2 Report

The authors analyzed the microbial communities - bacterial and fungal - present in the rhizosphere of three Arabidopsis genotypes, each mutated by the absence of a pattern recognition receptors gene (efr1, fls2 and cerk1), compared to wild-type control.

The research led to the identification of significant differences in the b -diversity of rhizospheric bacterial communities associated with the control plants versus the mutants, but not in a-diversity. Suggesting that the FLS2 receptor could, mechanistically, modulate plant responses to the root-associated bacteria.

The article is well written and the research is well set up. The introduction provides a comprehensive analysis of the effect of host genotype in the root-associated microbiome. The discussion on the results is also exhaustive but I suggest the authors to broaden both the introduction and the discussion / conclusions by reporting some considerations on the importance of this type of study to understand and better exploit the microbiome associated with plants in agriculture.

This study is limited to the rhizosphere community; it would also be interesting to address it to the large microbial community present within the plant.

Author Response

The authors analyzed the microbial communities - bacterial and fungal - present in the rhizosphere of three Arabidopsis genotypes, each mutated by the absence of a pattern recognition receptors gene (efr1, fls2 and cerk1), compared to wild-type control.

The research led to the identification of significant differences in the b -diversity of rhizospheric bacterial communities associated with the control plants versus the mutants, but not in a-diversity. Suggesting that the FLS2 receptor could, mechanistically, modulate plant responses to the root-associated bacteria.

The article is well written and the research is well set up. The introduction provides a comprehensive analysis of the effect of host genotype in the root-associated microbiome.

The discussion on the results is also exhaustive but I suggest the authors to broaden both the introduction and the discussion / conclusions by reporting some considerations on the importance of this type of study to understand and better exploit the microbiome associated with plants in agriculture. This study is limited to the rhizosphere community; it would also be interesting to address it to the large microbial community present within the plant.  

Done. Added new paragraph at the end of discussion and 4th paragraph at introduction stating potential biotechnological implications of better understanding plant host genotype and the microbiome interaction.